# A Reduction to Binary Approach for Debiasing Multiclass Datasets

**Ibrahim Alabdulmohsin**
Google Research
Zürich, Switzerland
ibomohsin@google.com

**Jessica Schrouff** *
Google Research
London, United Kingdom
schrouff@google.com

**Oluwasanmi Koyejo**
Google Research
Mountain View, United States
sanmik@google.com

## Abstract

We propose a novel reduction-to-binary (R2B) approach that enforces demographic parity for multiclass classification with non-binary sensitive attributes via a reduction to a sequence of binary debiasing tasks. We prove that R2B satisfies optimality and bias guarantees and demonstrate empirically that it can lead to an improvement over two baselines: (1) treating multiclass problems as multi-label by debiasing labels independently and (2) transforming the features instead of the labels. Surprisingly, we also demonstrate that independent label debiasing yields competitive results in most (but not all) settings. We validate these conclusions on synthetic and real-world datasets from social science, computer vision, and healthcare.

## 1 Introduction

Several studies have demonstrated that predictors are susceptible to unintended bias – for example, deep neural networks (DNNs) can amplify spurious correlations in the training data (Hendricks et al., 2018; Bolukbasi et al., 2016; Caliskan et al., 2017; Zhao et al., 2017; Yang et al., 2020b; Wang et al., 2020b; Stock and Cisse, 2018). Moreover, error disparities can arise, where the performance of the model for minority groups is disproportionately worse than for the majority (Zhao et al., 2017; Buolamwini and Gebru, 2018; Deuschel et al., 2020). Studies, such as Buolamwini and Gebru (2018) and Wang et al. (2020a), observe that one source of this disparity is that datasets may reflect societal stereotypes, thereby highlighting the importance of debiasing datasets.

Nevertheless, despite the proliferation of research on algorithmic fairness in recent years, very few methods exist that can handle multiclass classification tasks with non-binary sensitive attributes. This gap is particularly noteworthy given that this setting is the norm in real-world applications, not the exception. Considering multiclass classification, out of the 70+ image classification datasets in the TensorFlow Dataset catalog (Abadi et al., 2015), less than 10% are for binary classification problems. Along similar lines, considering non-binary sensitive attributes, the seven protected attributes according to the US Equal Credit Opportunity Act of 1974 (U.S. Government Publishing Office, 1974) are non-binary, such as gender, race, religion, and age.

Two approaches are available to handle such a broad setting, to the best of our knowledge. One option is to view the multiclass problem as multi-label and debias every label separately, i.e. transform the labels such that they are uncorrelated with the sensitive attribute, before normalizing the output into a

---

* Now at DeepMind

$$s = \begin{bmatrix} 0 \\ 1 \\ 0 \end{bmatrix}, \qquad Y = \begin{bmatrix} \frac{1}{2} & \frac{1}{2} & 0 \\ 1 & 0 & 0 \\ 0 & 0 & 1 \end{bmatrix} \xrightarrow{\text{debias}} \hat{Y} = \begin{bmatrix} 1 & \frac{1}{2} & 0 \\ \frac{1}{2} & \frac{1}{4} & \frac{1}{2} \\ 0 & 0 & 1 \end{bmatrix} \xrightarrow{\text{normalize}} \tilde{Y} = \begin{bmatrix} \frac{2}{3} & \frac{1}{3} & 0 \\ \frac{2}{5} & \frac{1}{5} & \frac{2}{5} \\ 0 & 0 & 1 \end{bmatrix}$$

Figure 1: An illustration of why treating multiclass problems as multi-label may not achieve demographic parity (DP). We assume a binary sensitive attribute $s$ and a matrix $Y$ of label conditional probability consisting of three data records (rows) and three classes (columns). Multiclass DP exists if $\mathbb{E}[\mathbf{y}|\mathbf{s}=0] \neq \mathbb{E}[\mathbf{y}|\mathbf{s}=1]$, where $\mathbf{y} \in \mathbb{R}^3$ a vector of probability scores over the three possible classes (see Equation 2). Observe that $Y$ does not satisfy DP because the probability scores are not independent of $\mathbf{s}$. The matrix $\hat{Y}$ (in middle) debiases every label separately (to see this, average $s=0$ rows and compare to the $\mathbf{s}=1$ row). But, to construct proper multiclass scores, its rows are normalized into probability distributions in $\tilde{Y}$, which reintroduces bias. See Appendix A for details.

valid probability distribution. However, this approach lacks fairness guarantees since normalizing the output can re-introduce bias (see Figure 1 for a cartoon illustration). Second, one can debias the instance *features* instead of the labels irrespective of the number of classes, such as using the demographic parity remover (Feldman et al., 2015), which debiases every feature separately in a rank-preserving manner. However, debiasing the features can be suboptimal when the number of features is large as we demonstrate in Section 4. Note, in particular, that the latter approach requires a good estimate of the true cumulative distribution function (CDF) for each feature, but the uniform rate of convergence of the multivariate empirical CDF to the true CDF decreases as the number of features grows (Naaman, 2021). We explore the performance of this approach and illustrate in Appendix B how it can fail when using, for example, equal-width binning to estimate the CDFs.

To address this gap in the literature, we propose a reduction-to-binary (R2B) approach for debiasing multiclass datasets that can accommodate an arbitrary number of classes and groups and does not require access to the sensitive attribute at prediction time. The proposed algorithm is based on the alternating direction method of the multipliers (ADMM) (Boyd et al., 2011), which is a framework for decomposing optimization problems into a sequence of parallel tasks. Using ADMM, we show that the task of debiasing multiclass datasets reduces to a sequence of parallel debiasing jobs on each class separately, along with an aggregation step. Each debiasing job can be executed using the randomized threshold optimizer (RTO) algorithm of Alabdulmohsin and Lucic (2021), which was originally proposed as a post-hoc rule for binary classification – in this work, we adopt it as a preprocessing method. The overall algorithm inherits the guarantees of ADMM including convergence and optimality. Our empirical results demonstrate that the proposed algorithm can lead to an improvement over the two outlined baselines; i.e. treating multiclass problems as multi-label, and transforming the features instead of the labels.

Surprisingly, we also demonstrate that the baseline multi-label approach yields competitive results in most (but not all) settings despite the potential impact of normalization on bias (cf. Figure 1). In Appendix A, we provide an argument for why this can happen under idealized assumptions.

**Statement of Contribution.** This work addresses a gap in the machine learning literature on demographic parity for multiclass classification with non-binary sensitive attributes. Our contributions are: (1) we derive a method for debiasing multiclass datasets with categorical sensitive attributes of arbitrary cardinality with respect to demographic parity – our approach reduces to a sequence of debiasing tasks on binary labels, (2) we establish theoretical guarantees for the proposed algorithm, (3) we study the impact of the experiment settings (e.g. number of features, number of classes, etc) on the debiasing algorithms using synthetic data and validate different debiasing methods on real-world datasets from three domains: social science, computer vision, and healthcare, and (4) we evaluate the baseline multi-label approach in debiasing multiclass datasets and demonstrate that it yields competitive results in most (but not all) settings.

## 2   Related Work

In the *binary* classification setting, several algorithms have been proposed for mitigating bias in machine learning. These are often classified into three groups depending on which step in the machine learning pipeline the debiasing effort takes place. First, there are *preprocessing* methods that are applied prior to training, such as by learning a fair representation (Zemel et al., 2013; Lum and

Johndrow, 2016; Bolukbasi et al., 2016; Calmon et al., 2017; Madras et al., 2018) or re-weighting training examples (Kamiran and Calders, 2012). One example of a preprocessing method that transforms the labels is the optimized score transformation (OST) method (Wei et al., 2019) whereas the DP remover method of Feldman et al. (2015) transforms the features. Feature debiasing has the advantage of not depending on the labels and can therefore be applied in any task setting.

Second, *in-processing* methods intervene during training, such as by adjusting the gradient updates (Zhang et al., 2018) or by adding explicit constraints into the optimization problem; e.g. (Zafar et al., 2019). In Agarwal et al. (2018), it is shown that many fairness criteria can be enforced during training via a reduction approach to cost-sensitive classification rules. Our proposed algorithm for the multiclass setting also reduces to a sequence of debiasing rounds. However, every round decomposes into multiple parallel debiasing jobs for each label separately (i.e. using algorithms for debiasing binary labels) and we operate in the preprocessing setting. Reduction methods, in which solutions to simple problems are reused to solve complex tasks, are not uncommon in machine learning. Besides Agarwal et al. (2018), other examples include the *MetaCost* method (Domingos, 1999), error correcting codes (Dietterich and Bakiri, 1994), boosting (Schapire and Freund, 2014), conditional probability estimation (Beygelzimer et al., 2009), ranking (Balcan et al., 2007), and relating reinforcement learning to classification (Langford and Zadrozny, 2005).

Third, many post-processing methods have been proposed in the literature (Corbett-Davies et al., 2017; Menon and Williamson, 2018; Celis et al., 2019; Kamiran et al., 2012; Hardt et al., 2016; Wei et al., 2019; Alabdulmohsin and Lucic, 2021). One recent example is the randomized threshold optimizer (RTO) of Alabdulmohsin and Lucic (2021), which can provably approximate the optimal unbiased predictor (i.e. it is statistically consistent) and can be solved efficiently via stochastic gradient descent (SGD). Our ADMM based approach for debiasing multi-class datasets utilizes RTO to debias every label separately before aggregating results.

To our knowledge, two methods have been developed for multiclass classifications. First is the algorithm of Denis et al. (2021), which assumes that the sensitive attribute is binary, whereas the sensitive attribute can be non-binary in our setting. Second, Yang et al. (2020a) propose both in-processing and post-processing procedures; the latter of which requires access to the sensitive attribute at prediction time and cannot be directly applied as a preprocessing method on discrete labels.

**Advantages of R2B.** The proposed ADMM-based reduce-to-binary (R2B) algorithm is a pre-processing method. This offers three immediate advantages. First, it is agnostic to the choice of the training algorithm; unlike, for example, in-processing methods that are often designed with a specific model and a choice of training method in mind. Second, R2B does not require access to the sensitive attribute at prediction time, which is a critical advantage over post-processing methods (Zafar et al., 2019). Third, R2B reduces the debiasing task to a sequence of rounds of *debiasing* the labels, not training classifiers. The computational overhead in our approach is often negligible compared to in-processing methods that provide a reduction approach to a sequence of classification rules, such as in Agarwal et al. (2018), in which the entire model is *trained* several times. Unlike the multi-label approach, R2B is guaranteed to debias the dataset up to the prescribed bias tolerance level. In addition, R2B performs better than other baselines, such as preprocessing the features instead of the labels. Importantly, it can be applied in the multiclass setting with a non-binary sensitive attribute.

## 3 Reduction to Binary Method

**Notation.** We reserve boldface letters for random variables (e.g. $\mathbf{x}$), small letters for instances (e.g. $x$), capital letters for matrices (e.g. $X$), and calligraphic typeface for sets (e.g. the instance space $\mathcal{X}$). If $f : \mathcal{X} \to \mathbb{R}^n$ is a multivariate function, we write $f_k(x) : \mathcal{X} \to \mathbb{R}$ for the $k$-th component of $f$. We denote $[n] = \{0, 1, \ldots, n-1\}$ and reserve $\eta_k(x)$ for the Bayes regressor: $\eta_k(x) = p(\mathbf{y} = k | \mathbf{x} = x)$. We assume that the sensitive attribute is known at training time and it has a finite range. We denote the instance space $\mathcal{X}$, the dataset $\mathcal{D} : |\mathcal{D}| = N$, the target set $\mathcal{Y} = \{0, 1, \ldots, L-1\}$ and the sensitive attribute $g : \mathcal{X} \to \mathcal{S}$ where $\mathcal{S} = [R]$. We write $\mathcal{X}_\mathcal{S}$ for the portioning of $\mathcal{X}$ induced by $\mathcal{S}$; i.e. $\mathcal{X}_\mathcal{S} = \{\mathcal{X}_0, \ldots, \mathcal{X}_{R-1}\}$ is the set of groups/subpopulations where $\mathcal{X} = \cup_{s=0}^{R-1} \mathcal{X}_s$. We will occasionally write $x_i$ for the $i$-th instance in the training dataset. We also denote $\mathbf{1} \in \mathbb{R}^n$ for the vector of all 1's and $\mathbf{0} \in \mathbb{R}^m$ for the vector of all zeros. In both cases, the dimension is implicit and should be inferred with ease from the context. Finally, $||X||_F$ is the Frobenius norm of the matrix $X$.

## 3.1 Multiclass Demographic Parity

**Definition.** Before deriving the reduction-to-binary (R2B) method, we first describe how demographic parity is extended to the multiclass setting. In the binary classification setting, demographic parity measures the difference in *mean* outcomes across groups. More precisely, let $f : \mathcal{X} \to [0, 1]$ be a binary predictor that outputs a probability score $f(x) = p(\mathbf{y} = 1 | \mathbf{x} = x)$. Then, $f$ is said to satisfy $\epsilon$ demographic parity if the following holds (Dwork et al., 2012; Zafar et al., 2017; Mehrabi et al., 2019; Alabdulmohsin and Lucic, 2021):

$$\max_{s \in \mathcal{S}} \mathbb{E}_{\mathbf{x}}[f(\mathbf{x}) \, | \, g(\mathbf{x}) = s] \; - \; \min_{s \in \mathcal{S}} \mathbb{E}_{\mathbf{x}}[f(\mathbf{x}) \, | \, g(\mathbf{x}) = s] \leq \epsilon, \tag{1}$$

where $g : \mathcal{X} \to \mathcal{S}$ is the sensitive attribute. A crowd-sourcing study found that DP matches with the common perception of bias (Srivastava et al., 2019). In the multiclass setting, let $f_k : \mathcal{X} \to [0, 1]$ be the probability score assigned to the class $k \in \mathcal{Y}$ given the instance $x \in \mathcal{X}$ and let $f : \mathcal{X} \to [0, 1]^L$ be the multivariate function $f(x) = (f_0(x), f_1(x), \ldots, f_{L-1}(x))^T$. In this work, we follow the definition used in Denis et al. (2021) and say that $f$ satisfies $\epsilon$ demographic parity if:

$$DP(f) \doteq \max_{k \in \mathcal{Y}} \left\{ \max_{s \in \mathcal{S}} \mathbb{E}_{\mathbf{x}}[f_k(\mathbf{x}) \, | \, g(\mathbf{x}) = s] \; - \; \min_{s \in \mathcal{S}} \mathbb{E}_{\mathbf{x}}[f_k(\mathbf{x}) \, | \, g(\mathbf{x}) = s] \right\} \leq \epsilon. \tag{2}$$

Hence, predictors with small demographic parity have similar mean outcomes across all groups $\mathcal{S}$ with respect to *all* of the predicted targets. We take the maximum here instead of the average to avoid pitfalls that can arise when some classes are more preferred than others, which is analogous to the "fairness gerrymandering" phenomenon, where predictors may exhibit small bias across the intersection of groups *on average*, but not at the worst-case intersection (Kearns et al., 2018). Here, instead of disparities across groups, we highlight worst-case disparities across *classes*. In Appendix C, we discuss some scaling effects of the number of classes on the definition of multiclass DP.

## 3.2 Derivation of the Reduction-to-Binary (R2B) Algorithm

In the binary classification setting where $\mathcal{Y} = \{0, 1\}$, a natural measure of performance is the 0-1 misclassification error rate. Minimizing the expected 0-1 error is equivalent to maximizing the linear functional $\mathbb{E}[\hat{y}(\mathbf{x})(2\eta_1(\mathbf{x}) - 1)]$ (Alabdulmohsin and Lucic, 2021), where $\hat{y}$ are the new debiased labels and $\eta_1(x) = p(\mathbf{y} = 1 | \mathbf{x} = x)$ is the Bayes regressor. Here, the minimization is with respect to the debiased labels $\hat{y} : \mathcal{X} \to \mathbb{R}^L$. Hence, if access to the Bayes regressor is available at training time, one can usually compute $\hat{y}(\mathbf{s})$ that minimizes the expected 0-1 loss subject to the desired fairness constraints by solving a convex optimization problem; cf. (Menon and Williamson, 2018; Alabdulmohsin and Lucic, 2021; Celis et al., 2019; Wei et al., 2019).

Unfortunately, such an advantage no longer holds in the multiclass setting because the top-1 accuracy is often insufficient. For top-k accuracy, however, the squared loss is statistically consistent (Yang and Koyejo, 2020). Hence, we propose to minimize the $\ell_2$ distance $\mathbb{E}[(\hat{y}(\mathbf{x}) - y(\mathbf{x}))^2]$ between the debiased predictions and the original labels. Throughout the sequel, $y(x) : \mathcal{X} \to \mathbb{R}^L$ gives the probability scores assigned to the different classes for the instance $x \in \mathcal{X}$, which can be a degenerate distribution as is often the case. The objective function $\mathbb{E}[(\hat{y}(\mathbf{x}) - y(\mathbf{x})^2)]$ can be written as:

$$(\lambda/2) \, \mathbb{E}_{\mathbf{x}} \, ||\hat{y}(\mathbf{x})||_2^2 \; - \; \mathbb{E}_{\mathbf{x}}[\hat{y}(\mathbf{x})^T y(\mathbf{x})] \; + \; \text{Constant}, \tag{3}$$

where $\lambda = 1$. Rewriting the objective function in the form (3) provides an alternative interpretation: by minimizing the objective in (3), one seeks a solution that minimizes the *regularized* top-1 error. This holds because the probability of correctly classifying an example $x$ equals $\hat{y}(x)^T y(x)$ when $y_k(x) = p(\mathbf{y} = k | \mathbf{x} = x)$. Regularization, however, discourages extreme predictions. We fix $\lambda = 1$ since it corresponds to minimizing the squared loss, although R2B can handle any $\lambda > 0$.

The task of debiasing labels in a dataset $\mathcal{D}$ can hence be cast into the convex optimization problem:

$$\min_{\hat{y}:\mathcal{X}\to\mathbb{R}^L} \quad \sum_{x \in \mathcal{D}} \left\{ (1/2) \, ||\hat{y}(x)||_2^2 \; - \; \hat{y}(x)^T y(x) \right\}. \tag{4}$$

$$\text{s.t.} \quad \max_{k \in \mathcal{Y}} \left\{ \max_{s \in \mathcal{S}} \mathbb{E}_{\mathbf{x}}[\hat{y}_k(\mathbf{x}) \, | \, g(\mathbf{x}) = s] \; - \; \min_{s \in \mathcal{S}} \mathbb{E}_{\mathbf{x}}[\hat{y}_k(\mathbf{x}) \, | \, g(\mathbf{x}) = s] \right\} \leq \epsilon.$$

$$\forall x \in \mathcal{D} : \; \mathbf{1}^T \hat{y}(x) = 1 \quad \wedge \quad \hat{y}(x) \geq \mathbf{0}.$$

**Algorithm 1:** Pseudocode of the reduction-to-binary (R2B) algorithm for debiasing multiclass datasets with categorical sensitive attributes of arbitrary cardinality.

---

**Inputs:** (1) Step size $\tau > 0$; (2) Demographic parity tolerance level $\epsilon \geq 0$; (3) Label matrix $Y \in \mathbb{R}^{N \times L}$, where $Y_{ik} = p(\mathbf{y} = k)$ for the $i$-th training example; (4) Sensitive attribute $g \in [R]^N$, where $g(i)$ is the sensitive class of the $i$-th training example.

**Output:** Debiased probability scores $\hat{Y} \in \mathbb{R}^{N \times L} : \hat{Y} \geq \mathbf{0} \wedge \hat{Y}\mathbf{1} = \mathbf{1}$ that minimize $||Y - \hat{Y}||_F$ subject to $\epsilon$ demographic parity; i.e. a solution to Equation (5).

**Training:** Set $Y^{(0)} = Y$ and $Z^{(0)} = U^{(0)} = \mathbf{0} \in \mathbb{R}^{N \times L}$. Repeat until stopping criterion (cf. Section 3.3):

1. *Debias in Parallel:*
   Set $F^{(t)} = Y + \tau\left(Z^{(t)} - U^{(t)}\right)$. Let $f_k^{(t)} \in \mathbb{R}^N$ be the $k$-th column of $F^{(t)}$. Solve the following debiasing task for each class separately (e.g. using RTO (Alabdulmohsin and Lucic, 2021)):

$$\hat{y}_k^{(t)} = \quad \arg\min_{\mathbf{0} \leq \hat{y}_k \leq \mathbf{1}} \quad (1+\tau)/2||\hat{y}_k||_2^2 - \hat{y}_k^T f_k^{(t)}$$

$$\text{s.t.} \quad \max_{s \in \mathcal{S}} \mathbb{E}_{\mathbf{i} \in [N]}[\hat{y}_k(\mathbf{i}) \mid g(\mathbf{i}) = s] - \min_{s \in \mathcal{S}} \mathbb{E}_{\mathbf{i} \in [N]}[\hat{y}_k(\mathbf{i}) \mid g(\mathbf{i}) = s] \leq \epsilon$$

2. *Aggregate Results:*
   (a) Set $H_{ki}^{(t+1)} = \hat{y}_k^{(t)}(i)$: unnormalized score assigned to class $k \in \mathcal{Y}$ for the $i$-th training example.
   (b) Normalize probability scores using:

$$Z^{(t+1)} = H^{(t+1)} + U^{(t)} - \frac{1}{L}\left[(H^{(t+1)} + U^{(t)}) \cdot \mathbf{1} - \mathbf{1}\right] \cdot \mathbf{1}^T.$$

   (c) Update: $U^{(t+1)} = U^{(t)} + H^{(t+1)} - Z^{(t+1)}$.

**Return** $\hat{Y} = Z$.

---

Instead of solving this optimization problem directly, we re-express it into a more convenient form:

$$\min_{\hat{y}:\mathcal{X}\to\mathbb{R}^L} \quad \sum_{x \in \mathcal{D}} \left\{ (1/2)\,||\hat{y}(x)||_2^2 \; - \; \hat{y}(x)^T y(x) \; + \; \mathbb{I}_{\mathcal{F}}(z(x)) \right\}. \tag{5}$$

$$\text{s.t.} \quad \max_{k \in \mathcal{Y}} \left\{ \max_{s \in \mathcal{S}} \mathbb{E}_{\mathbf{x}}[\hat{y}_k(\mathbf{x}) \mid g(\mathbf{x}) = s] - \min_{s \in \mathcal{S}} \mathbb{E}_{\mathbf{x}}[\hat{y}_k(\mathbf{x}) \mid g(\mathbf{x}) = s] \right\} \leq \epsilon.$$

$$\forall x \in \mathcal{D} : \; \hat{y}(x) = z(x) \quad \wedge \quad \mathbf{0} \leq \hat{y}(x) \leq \mathbf{1}.$$

We introduced a dummy constraint $\hat{y}(x) \leq \mathbf{1}$ and replaced $\mathbf{1}^T \hat{y}(x) = 1$ with a term $\mathbb{I}_{\mathcal{F}}(\hat{y}(x))$ in the objective function where:

$$\mathbb{I}_{\mathcal{F}}(\hat{y}(x)) = \begin{cases} 0, & \mathbf{1}^T\hat{y}(x) = 1 \\ \infty, & \text{otherwise.} \end{cases} \tag{6}$$

Clearly, the optimization problem (5) is equivalent to the original optimization problem (4).

Algorithm 1 presents pseudocode for the proposed reduction-to-binary (R2B) method for solving the optimization problem in (5). At a high-level, the algorithm decomposes the debiasing task into a sequence of rounds. In each round, the probability scores assigned to different classes are debiased *independently*. This produces a new set of scores, denoted $H^{(t)}$, that are not normalized. The next step is to normalize them so that they sum to one. The set of normalized probability scores is given by $Z^{(t)}$. After that, the objective function is altered slightly and the process is repeated. We prove that Algorithm 1 returns the *optimal* solution to (5) and discuss suitable stopping criteria in Section 3.3.

### 3.3 Analysis of the Algorithm

**Proposition 1** (Optimality). *Algorithm 1 terminates with an optimal solution to (5), in which the debiased probability scores assigned to the $i$-th example are given by the $i$-th row of the matrix $\hat{Y}$.*

**Stopping Criteria.** The simplest stopping criterion to use in Algorithm 1 is the number of ADMM rounds. Boyd et al. (2011) observes that 50-100 rounds are often sufficient, which we also observe to be true in our experiments (see Figure 3). Alternatively, one may use the sum of the "primal" and

"dual residuals". In Algorithm 1, the optimality conditions of ADMM (Boyd et al., 2011) reduce to the following conditions $Z^{(t)} = H^{(t)}$ and $Z^{(t+1)} = Z^{(t)}$. Hence, we may define $\delta_p = ||Z^{(t)} - H^{(t)}||_F$ and $\delta_r = ||Z^{(t+1)} - Z^{(t)}||_F$ and stop when $\delta_p + \delta_r$ are below a prescribed threshold. In our implementation, we follow the first approach and set the maximum number of ADMM rounds to 100.

**Bias Guarantee.** For our next result, we write $\hat{\mathbb{E}}[f(\mathbf{x})]$ for the average value of a function $f$ on the training examples, and write $\mathbb{E}[f(\mathbf{x})]$ for the expectation of $f$ over the true distribution of instances.

**Proposition 2** (Bound on bias). *Let $\mathbb{P}^L$ be the probability simplex in $\mathbb{R}^L$ and $\mathcal{F}$ be a class of functions from $\mathcal{X}$ to $\mathbb{P}^L$, such that the set $\{f_k : f \in \mathcal{F}\}$ has a Rademacher complexity $d$. Suppose that all training examples $(\mathbf{x}, \mathbf{y}, \mathbf{s}) \in \mathcal{D}$ are drawn i.i.d. and are debiased using Algorithm 1 with bias level $\epsilon \geq 0$. Let $h(\mathbf{s}) : \mathcal{X} \to \mathbb{P}^L$ be the debiased labels. Let $f \in \mathcal{F}$ be the final classifier. Then, with a probability of at least $1 - \delta$, we have: $DP(f) \leq \epsilon + 2\tau + 2d + \sqrt{\frac{\log \frac{2LR}{\delta}}{2n_0}}$, where $n_0 = \min_{s \in [R]} \left| \{\mathbb{I}\{g(\mathbf{x}) = s\} : \mathbf{x} \in \mathcal{D}\} \right|$ is the size of the smallest group in the training dataset, $L = |\mathcal{Y}|$ is the number of classes, $R = |\mathcal{S}|$ is the number of groups, and $\tau = \sup_{k \in \mathcal{Y}, s \in [R]} \left| \hat{\mathbb{E}}[f_k(\mathbf{x}) \,|\, g(\mathbf{x}) = s] - \hat{\mathbb{E}}[h_k(\mathbf{x}) \,|\, g(\mathbf{x}) = s] \right|$, where both expectations are measured on the training dataset.*

Proposition 2 provides a formal justification to the preprocessing approach; it states that if the training labels are debiased using Algorithm 1, then a classifier trained on the debiased data is guaranteed to exhibit small bias in the future as long as four conditions are satisfied: (1) the level of bias in the training data is small; i.e. $\epsilon \ll 1$ in Algorithm 1, (2) the classifier fits the training examples well; i.e. $\tau \ll 1$, (3) the complexity of the classifier is not too large (does not memorize examples); i.e. $d \ll 1$, and (4) there exists a large number of training examples for each group in $\mathcal{S}$; i.e. $n_0 \gg \log LR$.

## 4 Experiments

We compare R2B against the two baselines: (1) treating multiclass datasets as multi-label, and (2) transforming the features. In the multi-label approach, we use two recent algorithms for debiasing binary labels: optimized score transformation (OST) (Wei et al., 2019) and RTO (Alabdulmohsin and Lucic, 2021) with $\gamma = 0.01$ and $\rho = \mathbb{E}[\mathbf{y}_k]$ (default) values. After that, the scores are normalized to sum to one. Note that $\gamma \ll 1$ in the RTO algorithm corresponds approximately to a *hard*-thresholding rule, hence we refer to it as hard-threshold optimizer (HTO). In the second approach, we apply the demographic parity remover (DPR) method of Feldman et al. (2015), which debiases features. In the latter case, we use equal-mass binning when estimating the cumulative density function (see Appendix B). All three baselines can be motivated via provable optimality guarantees (Wei et al., 2019; Alabdulmohsin and Lucic, 2021; Feldman et al., 2015). We also include for comparison training without debiasing the data, which we denote as BL[2]. We note that existing multiclass fairness approaches such as Yang et al. (2020a) cannot be applied for pre-processing as they require either probability estimates or weighted classifiers. We highlight in **boldface** the method with the best accuracy among the set of methods that achieve the smallest bias within the margin of error.

In R2B, on the other hand, we use a step size of $\tau = 0.5$ in Algorithm 1 and a maximum number of 100 ADMM rounds. We also report results when a *single* round of R2B is used, which we denoted as R2B$_0$. Except for the healthcare dataset whose splits are fixed, we split data at random into 25% for test and 75% for training. We debias the dataset prior to training and measure performance (e.g. accuracy and DP) on the test split. All methods use the same splits. We re-run every experiment with ten different random seeds and report both the averages and 99% confidence intervals.

### 4.1 Synthetic Dataset

We begin our analysis with synthetic data. In this dataset, let $L$ be the number of classes and $d$ be the number of features. We use a mixture of $L$ Gaussians whose means are sampled from $\mathcal{N}(0, \sigma^2 I)$ with $\sigma^2 = 1/d$ and $\mathcal{Y} = [L]$. For the sensitive attribute, we set it equal to $\mathbb{I}\{\mathbf{y} = 0\}$ with probability $1/2$; otherwise, it is chosen uniformly at random from the set $\{0, \ldots, 4\}$; i.e. $|\mathcal{S}| = 5$. Hence, bias is introduced into the dataset. The classifiers are $k$-NN with $k = 5$ and Random Forest (RF) with

---

[2]Source code is publicly available at: `https://github.com/google-research/google-research/tree/master/ml_debiaser`

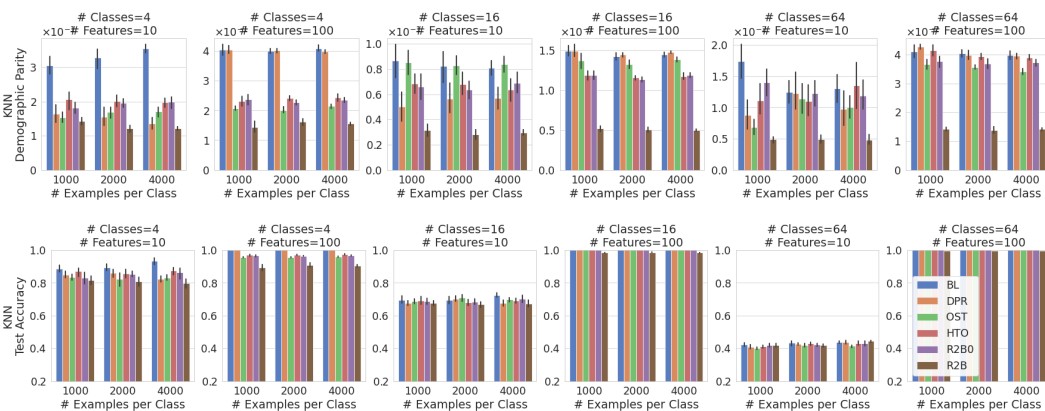

Figure 2: Top two displays the multiclass demographic parity (DP) as measured using (2) for the synthetic dataset with varying numbers of classes, features, and training examples using $k$NN as a classifier (see Appendix G for full figures). The reduction-to-binary (R2B) method provides a stronger fairness guarantee than the competing methods. The bottom row shows the prediction accuracy.

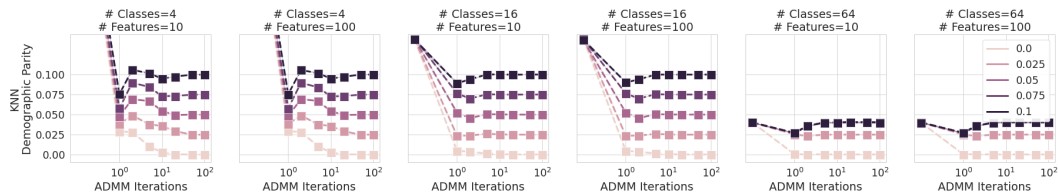

Figure 3: The level of bias in the *training* data is plotted as measured by (2) vs. the number of ADMM rounds for different levels of bias tolerance $\epsilon \in \{0, 0.025, 0.05, 0.075, 0.10\}$ (see legend) using the $k$NN classifier (see full figures in Appendix G). In general, 50-100 rounds of ADMM in R2B are sufficient to reach convergence in agreement with prior observations (Boyd et al., 2011).

maximum depth 5, both implemented using Scikit-Learn (Pedregosa et al., 2011). We vary the number of classes in $\{4, 16, 64\}$, the number of features in $\{10, 100\}$, and the number of training examples per class in $\{1000, 2000, 4000\}$. We set the bias tolerance $\epsilon$ to zero in all experiments since, otherwise, these methods provide incomparable mechanisms for controlling bias. Figure 2 displays the results for $k$-NN and Appendix G contains the full figures.

As shown in Figure 2, R2B provides a stronger fairness guarantee than the other competing methods. In addition, we observe that debiasing the features performs poorly when the number of features is large. As mentioned earlier, one possible explanation is that as the number of features grows, the rate of convergence of the multivariate empirical cumulative distribution function (EDF) to the true cumulative distribution function (CDF) decreases (Naaman, 2021). However, debiasing the features in Feldman et al. (2015) relies on having a correct estimate of the CDF since it matches quantiles.

Moreover, we observe that the multi-label approach (i.e. OST, HTO, and R2B0) can also fail in several cases. As illustrated in the cartoon example of Figure 1, this is because normalization impacts bias. In one experiment, for example, the level of DP in the training labels is reduced to less than 0.01 after debiasing labels separately using OST but bias increases to over 0.22 following normalization.

In Figure 3, we also plot the level of DP in the *training* labels as a function of the number of ADMM rounds in R2B for five different levels of bias tolerance $\epsilon \in \{0, 0.025, 0.05, 0.075, 0.10\}$. We observe that R2B converges in about 50 ADMM iterations, which is consistent with earlier observations in the literature (Boyd et al., 2011). In addition, when the level of bias in the data is smaller than $\epsilon$, the algorithm does not introduce any additional bias as expected.

Table 1: A summary of the performance results (99% confidence intervals) for the five debiasing algorithms on the Adult Income benchmark dataset, where the goal is to predict the education level. For each of the two classifiers Random Forest (RF) and $k$-NN, we either use the original data ($p = 0$) or inject bias ($p = 0.5$) as described in Section 4.2. First column is the bias in the data. The debiasing methods are the baseline (BL), DP remover (DPR) (Feldman et al., 2015), optimized score transformation (OST) (Wei et al., 2019), randomized threshold optimizer with $\gamma = 0.01$ (HTO) (Alabdulmohsin and Lucic, 2021), $R2B_0$, and R2B (our method).

| Criterion | $p$ | Learner | DATA | BL | DPR | OST | HTO | $R2B_0$ | R2B (ours) |
|---|---|---|---|---|---|---|---|---|---|
| DP [%] | 0.0 | RF | 5.0 | $3.8 \pm .2$ | $\mathbf{1.5 \pm .3}$ | $3.7 \pm .3$ | $2.9 \pm .3$ | $2.8 \pm .3$ | $2.2 \pm .2$ |
| | | KNN | 5.0 | $4.7 \pm .5$ | $3.0 \pm .7$ | $4.5 \pm .5$ | $3.2 \pm .5$ | $3.1 \pm .5$ | $\mathbf{2.1 \pm .4}$ |
| | 0.5 | RF | 48.0 | $22.8 \pm .3$ | $9.5 \pm .4$ | $1.7 \pm .3$ | $2.2 \pm .3$ | $2.1 \pm .2$ | $\mathbf{1.0 \pm .2}$ |
| | | KNN | 48.0 | $26.5 \pm .5$ | $23.3 \pm .6$ | $2.9 \pm .4$ | $3.4 \pm .6$ | $3.3 \pm .7$ | $\mathbf{1.1 \pm .2}$ |
| ACC [%] | 0.0 | RF | ★ | $42.6 \pm .6$ | $41.0 \pm .7$ | $42.6 \pm .6$ | $42.5 \pm .6$ | $42.6 \pm .5$ | $42.5 \pm .6$ |
| | | KNN | ★ | $35.5 \pm .5$ | $33.4 \pm .3$ | $35.3 \pm .5$ | $35.2 \pm .5$ | $35.2 \pm .5$ | $35.5 \pm .5$ |
| | 0.5 | RF | ★ | $42.0 \pm .7$ | $39.0 \pm .7$ | $34.1 \pm .4$ | $33.9 \pm .6$ | $34.0 \pm .5$ | $34.0 \pm .4$ |
| | | KNN | ★ | $34.9 \pm .5$ | $34.3 \pm .6$ | $30.2 \pm .4$ | $29.4 \pm .4$ | $29.6 \pm .7$ | $31.1 \pm .4$ |

## 4.2 Real-world Applications

Next, we validate R2B on applications from three domains: (1) social science, (2) computer vision, and (3) healthcare. Experiments involving neural networks are executed on Tensor Processing Units.

**Adult Income dataset.** The first dataset is the Adult Income dataset (Kohavi, 1996). In our case, we predict the education level of each individual from the remaining attributes. There are 16 classes in this dataset and 29 features, such as marital status, age, and occupation. The total number of examples (both training and test) is 48,842. The sensitive attribute is sex and only includes two categories ('Male', 'Female'). We compare the debiasing methods in Table 1 for both the Random Forest (RF) and $k$-NN classifiers (top two rows). Besides the original dataset, we also introduce bias and class imbalance into the data (both training and test) and compare methods: with probability $p = 0.5$, the label $\mathbf{y} \in \{0, 1, \ldots, 15\}$ is set to be equal to the sensitive attribute $\mathbf{s} \in \{0, 1\}$. This increases bias in the original labels from about 0.05 to around 0.48 and introduces class imbalance as well[3]. The performance of each debiasing method in this dataset is shown in Table 1, bottom two rows. In most cases, R2B provides a stronger bias guarantee compared to the other methods. Appendix F provides similar performance results for Top-K accuracy.

**COCO dataset.** The second dataset we use is the COCO dataset (Lin et al., 2014). It contains 80 classes corresponding to the objects in each image, such as chairs, cars, and handbags. We transform this multi-label problem into a multiclass problem using *soft* labels: we set the target label $\mathbf{y}$ to be equal to the *fraction* of the objects that belong to each class in each image. The soft label approach corresponds formally to the task of predicting the *distribution* of objects seen in the image or, equivalently, the task of predicting the class of an object chosen uniformly at random from the corresponding image. We follow the procedure of (Wang et al., 2020a) in inferring the sensitive attribute based on the image caption: we use images that contain either the word "woman" or the word "man" in their captions but not both. The total number of examples is 22,616. Because the task here is to predict the distribution of classes seen in the image, we define accuracy in terms of the total variation distance between probability distributions. Specifically, let $\hat{\mathbf{y}}$ be the model's prediction, then $\mathrm{acc}(\hat{\mathbf{y}}, \mathbf{y}) = 1 - (1/2)||\hat{\mathbf{y}} - \mathbf{y}||_1$. Note that $0 \leq \mathrm{acc}(\hat{\mathbf{y}}, \mathbf{y}) \leq 1$. See Appendix H for further details about the training procedure. Besides the original dataset, we also introduce bias and class imbalance with $p = 0.5$ as described previously. Results are provided in Table 2. As shown in the table, both R2B and the multi-label methods perform much better than transforming the features (DPR), which is consistent with the earlier observations on synthetic data. Moreover, R2B performs better overall.

**Dermatology.** In this task, we are interested in predicting 27 skin conditions (26 plus an 'other' label representing the long tail) from images of the pathology of interest, the patient's age and sex[4].

---

[3] The level of demographic parity in the original data can be different from the baseline results, because the baseline model may not predict the original labels. For instance, when $p = 0.5$, noise is added to the labels to make them biased but such noise can be ignored by the model.

[4] Sex corresponds to clinician or self recorded sex and only includes two categories ('Male', 'Female')

Table 2: A summary of the performance results (99% confidence intervals) for the five debiasing algorithms on the COCO benchmark dataset with soft labels. See Section 4.2 for details. Similar to Table 1, we experiment with both the original dataset ($p = 0$) and with introduced bias ($p = 0.5$).

|  | $p$ | DATA | BL | DPR | OST | HTO | $R2B_0$ | R2B (ours) |
|---|---|---|---|---|---|---|---|---|
| DP [%] | 0.0 | 6.2 | $2.6 \pm .4$ | $6.4 \pm 1.8$ | $2.2 \pm .3$ | $3.8 \pm .7$ | $2.2 \pm .4$ | $\mathbf{2.4 \pm .4}$ |
|  | 0.5 | 52.9 | $4.0 \pm .6$ | $52.3 \pm 3.4$ | $0.7 \pm .2$ | $1.3 \pm .2$ | $1.0 \pm .4$ | $\mathbf{1.1 \pm .4}$ |
| ACC [%] | 0.0 | $\star$ | $40.4 \pm .2$ | $28.0 \pm 2.0$ | $39.8 \pm .2$ | $32.8 \pm .1$ | $40.6 \pm .2$ | $\mathbf{41.3 \pm .2}$ |
|  | 0.5 | $\star$ | $43.8 \pm .3$ | $50.1 \pm 2.7$ | $42.5 \pm .2$ | $39.6 \pm .2$ | $43.8 \pm .2$ | $\mathbf{48.1 \pm .4}$ |

Table 3: A summary of performance results on a Dermatology dataset (Liu et al., 2020), where the goal is to predict the clinical condition from images of the pathology. DP is 0.09 in the original data.

| Metric | BL | DPR | OST | HTO | $R2B_0$ | R2B (ours) |
|---|---|---|---|---|---|---|
| DP [%] | $11.1 \pm 1.2$ | $14.3 \pm 2.2$ | $\mathbf{5.1 \pm 1.0}$ | $8.0 \pm 1.4$ | $\mathbf{4.8 \pm 0.5}$ | $\mathbf{5.0 \pm 0.5}$ |
| Top-1 [%] | $58.6 \pm 0.5$ | $58.3 \pm 0.7$ | $\mathbf{58.5 \pm 0.5}$ | $48.9 \pm 0.9$ | $\mathbf{58.3 \pm 0.7}$ | $\mathbf{59.1 \pm 0.6}$ |
| Top-2 [%] | $\mathbf{79.1 \pm 0.7}$ | $78.6 \pm 0.5$ | $\mathbf{79.3 \pm 0.6}$ | $67.5 \pm 0.3$ | $\mathbf{79.0 \pm 0.1}$ | $\mathbf{79.0 \pm 0.1}$ |
| Top-3 [%] | $89.1 \pm 0.4$ | $88.8 \pm 0.6$ | $\mathbf{88.8 \pm 0.6}$ | $78.4 \pm 0.8$ | $87.2 \pm 0.6$ | $87.2 \pm 0.6$ |

The dataset is a subset of the one used in Liu et al. (2020) and consists of de-identified retrospective adult cases from a teledermatology service serving 17 sites from 2 states in the United States. It is split according to condition prevalence for training ($n = 12,024$), tuning for hyper-parameters ($n = 1,925$) and hold-out testing ($n = 1,924$). We train a deep learning model to predict skin conditions as a multiclass task, as per the architecture described in Liu et al. (2020); Roy et al. (2021). For debiasing, we fit R2B on the tune split and assess the model performance. Following Liu et al. (2020), we report the top-1, top-2, and top-3 accuracy on the test split. Table 3 reports DP on the test split as well. For ease, we bucket age according to [18,30], [30,45], [45, 65] and [65,90) for debiasing. We note that the number of cases per condition and per intersection of the attributes might be low, as is common in data-scarce domains such as healthcare. Results are presented in Table 3.

**Summary of Findings.** The reduction-to-binary (R2B) approach performs at least as well as the other baselines in all of the experiments and outperforms the other methods in some cases, such as in the synthetic data, the Adult Income dataset, and COCO. In addition, the multi-label approach using OST and $R2B_0$ seem to offer competitive results in most (but not all) settings. In Appendix A, we provide an argument for why this might happen under idealized assumptions. However, it is worth emphasizing that the multi-label approach is not guaranteed to debias datasets successfully as illustrated earlier in Figure 1 and demonstrated using synthetic (Figure 2) and real (Table 1) data. On other other hand, R2B offers strong guarantees (cf. Propositions 1 and 2).

## 5 Discussion and Limitations

In this paper, we derive a reduction approach for debiasing multiclass datasets according to demographic parity (DP). The algorithm reduces the overall task into a sequence of parallel debiasing jobs

Table 4: A summary of the observed *error parities* for the five debiasing algorithms on all of the classification tasks, which is defined to be the difference between the maximum and minimum losses conditioned on each group. In the original datasets (i.e. $p = 0$), we do not observe an increase in error parity when the training data is debiased to account for DP. However, error parity seems to increase when introducing large bias to such data (i.e. with $p = 0.5$).

| p | Task | BL | DPR | OST | HTO | $R2B_0$ | R2B (ours) |
|---|---|---|---|---|---|---|---|
| 0 | Adult (RF) | $0.7 \pm 0.1$ | $0.7 \pm 0.4$ | $1.0 \pm 0.2$ | $0.8 \pm 0.2$ | $1.0 \pm 0.2$ | $0.7 \pm 0.3$ |
|  | Adult (KNN) | $0.7 \pm 0.3$ | $0.7 \pm 0.2$ | $0.7 \pm 0.3$ | $0.8 \pm 0.3$ | $0.8 \pm 0.3$ | $0.7 \pm 0.3$ |
|  | COCO | $4.4 \pm 0.5$ | $5.7 \pm 1.0$ | $4.0 \pm 0.5$ | $5.1 \pm 0.6$ | $4.2 \pm 0.5$ | $5.1 \pm 0.5$ |
|  | Derm | $5.6 \pm 1.9$ | $5.1 \pm 0.8$ | $4.7 \pm 1.9$ | $5.7 \pm 2.3$ | $6.1 \pm 1.8$ | $5.9 \pm 2.1$ |
| 0.5 | Adult (RF) | $7.6 \pm 1.1$ | $31.5 \pm 0.8$ | $49.9 \pm 0.2$ | $50.1 \pm 0.3$ | $49.9 \pm 0.4$ | $49.8 \pm 0.3$ |
|  | Adult (KNN) | $2.0 \pm 0.5$ | $1.2 \pm 0.4$ | $12.5 \pm 0.8$ | $8.1 \pm 0.3$ | $7.8 \pm 0.5$ | $31.3 \pm 0.7$ |
|  | COCO | $23.1 \pm 0.7$ | $25.4 \pm 3.0$ | $21.9 \pm 0.4$ | $17.1 \pm 0.4$ | $22.7 \pm 0.5$ | $38.7 \pm 0.8$ |

on binary labels. Due to this reduction, the algorithm scales well to large datasets with several classes. We verify empirically on both synthetic and real-world datasets that it outperforms other baselines.

Nevertheless, it is worth emphasizing that "fairness" is a societal concept and should not be reduced to statistical metrics, such as DP (Dixon et al., 2018; Selbst et al., 2019). As such, the claims of this paper hold for the narrow technical definition of DP, not for fairness in its broader sense.

One limitation in R2B is that it accommodates DP but cannot accommodate error parity metrics, such as equalized odds. This is because it operates in the pre-processing setting, where prediction "errors" are yet undefined. In particular, for anti-causal predictive tasks (Schölkopf et al., 2012) as is the case in dermatology (Castro and Glocker, 2020), Veitch et al. (2021) suggest that equalized odds would be a causally-grounded fairness constraint. Nevertheless, in real-world settings, we do not observe an increase in error parity when the training data is debiased to account for DP as shown in Table 4, except when bias is *introduced* into the data (i.e. with $p = 0.5$)). Mitigating DP can reduce accuracy when the labels are correlated with the sensitive attribute as has been noted in several works; e.g. (Menon and Williamson, 2018; Zhao and Gordon, 2019). Finally, it accommodates *categorical* sensitive attributes only, not continuous. While the majority of protected attributes are indeed categorical and continuous attributes can be converted into categorical features by binning their values (e.g. age in the dermatology example), handling continuous sensitive attributes without resorting to binning remains a challenge. We leave such questions for future work.

## Acknowledgements

The authors would like to acknowledge and thank Yuan Liu and Alex Brown at Google Health AI for their support with the dermatology application, and thank Kathy Meier-Hellstern and Lucas Dixon from Google Responsible AI team for their feedback on earlier drafts of this manuscript.

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
