# Appendix

## A The Multi-label approach and Bias

### A.1 Why can the multi-label approach fail?

We describe the cartoon illustration in Figure 1. Suppose we have a binary sensitive attribute $\mathbf{s}$ and three classes. Suppose we have three data records. The first record has the true label probability score $\mathbf{y} = (\frac{1}{2}, \frac{1}{2}, 0)$ as shown in the first row of the matrix $Y$ in the figure. The sensitive attribute of the first record is $\mathbf{s} = 0$, as shown in the first row of $\mathbf{s}$ in the figure. The same goes for the second and third records.

According to the original labels $Y$, $\mathbb{E}[\mathbf{y}|\mathbf{s} = 0]$ is the average of the first and third rows in $Y$, which is $(\frac{1}{4}, \frac{1}{4}, \frac{1}{2})$. On the other hand, $\mathbb{E}[\mathbf{y}|\mathbf{s} = 1]$ is equal to the second row of $Y$, which is $(1, 0, 0)$. So, demographic parity exists in the original labels because $\mathbb{E}[\mathbf{y}|\mathbf{s} = 0] \neq \mathbb{E}[\mathbf{y}|\mathbf{s} = 1]$. Using the definition of multiclass bias in (2), the level of bias in this case is $3/4$, where the maximum difference occurs in the first label.

The matrix $\hat{Y}$, on the other hand, is a transformation of the original labels $Y$ that is unbiased. Note that its second row equals the average of the first and last rows. However, it does not provide valid predictions because its rows are not normalized. This corresponds to the multi-label approach.

After normalization, bias is re-introduced again in the matrix $\tilde{Y}$. We have $\mathbb{E}[\tilde{\mathbf{y}}|\mathbf{s} = 0] = (\frac{1}{3}, \frac{1}{6}, \frac{1}{2})$ and $\mathbb{E}[\tilde{\mathbf{y}}|\mathbf{s} = 1] = (\frac{2}{5}, \frac{1}{5}, \frac{2}{5})$. The level of bias according to the definition in (2) is $0.1$, which indeed decreases but remains, nevertheless, far from the prescribed level. This failure does not only occur in contrived settings. As demonstrated in the synthetic data and real-world applications, the multi-label approach can indeed be sub-optimal in practice.

### A.2 When does the multi-label approach succeed?

In the previous section, we showed that the multi-label approach can fail in some settings. However, it seems to perform well in most of the experiments. We present an argument for why this is the case, next.

Let us consider one subpopulation only for now (e.g. $\mathbf{s} = 0$). Suppose we have $n$ examples and $L$ classes. Suppose that the labels after the multi-label debiasing step (but before normalization) are encoded in a matrix $H \in \mathbb{R}^{n \times L}$, whose rows give the probability of each class but they are not normalized. One way to normalize the rows is using:

$$Z = H - \frac{1}{L}[H \cdot \mathbf{1} - \mathbf{1}] \cdot \mathbf{1}^T$$

This is the normalization that minimizes the Frobenius norm $||H - Z||_F$ as shown in the proof of Proposition 1. Now, let's consider the average predictions across the columns:

$$\frac{1}{n} \mathbf{1}^T Z = \frac{1}{n} \left( \mathbf{1}^T H - \frac{1}{L} [\mathbf{1}^T H \mathbf{1} - n] \cdot \mathbf{1}^T \right)$$

Let $Z_k$ be the $k$-th column of $Z$ and similarly for $H$. The change in the average predictions before and after normalization satisfies:

$$\max_{k \in \{1,2,...,L\}} \left\{ \mathbf{1}^T (Z_k - H_k) \right\} \leq \frac{1}{n} ||\mathbf{1}^T (Z - H)||_1 = \frac{|\mathbf{1}^T H \mathbf{1} - n|}{n}$$

Now, suppose that $H = Y + E$, where $Y$ is the original matrix of labels (i.e. it is normalized) and $E \in [-1, 1]^{n \times L}$ is a random perturbation matrix that has a zero mean. In other words, we consider the case in which debiasing the labels separately behaves like a random perturbation to the original matrix $Y$. Then:

$$\max_{k \in \{1,2,...,L\}} \left\{ \mathbf{1}^T (Z_k - H_k) \right\} \leq \frac{|\mathbf{1}^T (Y + E) \mathbf{1} - n|}{n} = \frac{|\mathbf{1}^T E \mathbf{1}|}{n}.$$

However, $\mathbf{1}^T E \mathbf{1}/n$ is the average of $n$ random variables under our assumption that are bounded in the interval $[-L, +L]$. By the Hoeffding bound, we have:

$$p\left\{ \frac{|\mathbf{1}^T E \mathbf{1}|}{n} \geq \epsilon \right\} \leq 2 \exp\left\{ -\frac{\epsilon^2 n}{2L^2} \right\}$$

In our case, $H$ is a debiased matrix for a single subpopulation, which means that $\mathbb{E}[\mathbf{y} \,|\, \mathbf{s}] = \mathbf{1}^T H / n$ is the same for all groups. By the union bound, the probability that the average predictions do not deviate by more than $\delta$ in any single label is bounded by:

$$\sum_{s=1}^{R} 2 \exp\left\{ -\frac{\epsilon^2 n_s}{2L^2} \right\},$$

where $n_s$ is the number of training examples for the group $s$. This suggests that the multilabel approach is likely to work well when all of the groups have a large number of training examples available and if debiasing the labels separately behaves indeed like a random perturbation to the original label matrix.

## B   Transforming the features: equal-mass binning vs. equal-width

We use in our implementation of the DP remover algorithm of Feldman et al. (2015) *equal-mass* binning, instead of equal-width binning, when estimating the cumulative distribution function (CDF) of each feature. It has been noted that equal-mass binning seems to be a better estimator than equal-width binning Roelofs et al. (2020). We demonstrate below how equal-width binning using the Freedman–Diaconis rule Freedman and Diaconis (1981) can fail at eliminating bias in the DP remover algorithm.

Suppose that the sensitive attribute $\mathbf{s}$ is binary and let the distirbution of features $\mathbf{x} \in \mathbb{R}$ be given by:

$$p(\mathbf{x} = x \,|\, \mathbf{s} = 0) = \mathcal{N}(0, \epsilon^2)$$
$$p(\mathbf{x} = x \,|\, \mathbf{s} = 1) = \frac{1}{2}\mathcal{N}(0, \epsilon^2) + \frac{1}{2}\mathcal{N}(1, \epsilon^2)$$

Then, when $\epsilon \ll 1$, the interquantile range in the class $\mathbf{s} = 1$ is much larger than $\epsilon$. In the Freedman–Diaconis rule Freedman and Diaconis (1981), the bin width is proporitonal to the interquantile range. Hence, for any sample size $n$, there exists a constant $\epsilon \ll 1$ that is small enough for the entire data to fall into two bins only when $\mathbf{s} = 1$. The CDF function in the latter case is flat everywhere except at $\{0, 1/2, 1\}$.

On the other hand, the CDF using the same Freedman–Diaconis rule is a continuous function for the class $\mathbf{s} = 0$ (because it comprises of a single Gaussian density).

Now, consider the "median distribution" used in the DP remover algorithm. Let $F_s(\tau)^{-1}$ be the $\tau$ quantile of the features conditioned on $\mathbf{s} = s$. When, $\mathbf{s} = 1$, the feature value $\mathbf{x}$ will be mapped to one of three values only:

$$\left\{ \frac{F_0(0)^{-1} + F_1(0)^{-1}}{2}, \; \frac{F_0(0.5)^{-1} + F_1(0.5)^{-1}}{2}, \; \frac{F_0(1)^{-1} + F_1(1)^{-1}}{2} \right\}$$

When $\mathbf{s} = 0$, on the other hand, a feature value $x$ is mapped to $(F_0(\tau)^{-1} + F_1(\tau)^{-1})/2$, where $\tau = F_0(x)$. In the latter case, the range of the mapping is not restricted to three values only as in the former case. Hence, the two distributions will be different, and the features are no longer unbiased.

## C   Scaling Effects of the Number of Classes on the Multiclass DP Definition

The definition of demographic parity in (2) is computed over the underlying distribution of data. In practice, one measures bias on a *finite* sample. In the finite sample setting, even a perfectly unbiased predictor may appear to be biased according to (2) when the number of classes is large due to the nature of the distribution of extreme values. More precisely, let $L > 1$ be the number of classes and suppose that the classifier is unbiased; i.e. $\epsilon = 0$ with respect to the data distribution and assume that the central limit theorem holds when estimating the mean outcomes per group $\mathbb{E}_{\mathbf{x}}[f_k(\mathbf{x}) \,|\, g(x)]$. Let $\hat{\epsilon}$ be the *empirical* estimate of bias. Then, a well-known result states that $\hat{\epsilon} = \Theta(\sqrt{\log L})$ (Massart, 2007), which increases (albeit slowly) as the number of classes $L$ grows. This is one scaling effect of the number of classes $L$ on the definition of demographic parity.

Another scaling effect relates to *sparsity*: as the number of classes grows, the predictions $f_k(\mathbf{x})$ of a single label $k \in \mathcal{Y}$ become sparser. More precisely, since $\sum_k f_k(\mathbf{x}) = 1$, we have $\mathbb{E}_{\mathbf{x}}[f_k(\mathbf{x})] = 1/L$

on average. Due to this scaling effect, the level of demographic parity usually *decreases* as the number of classes increases, which agrees with the experimental results (see Figure 2).

A third scaling effect is class imbalance. Given $L$ classes and $n$ training examples, consider the case in which all classes are equally probable; i.e. $p(\mathbf{y} = k) = 1/L$ for all $k \in [L]$. Then, class imbalance can arise even in this setting due to the random sampling of training examples. For instance, if $n_k$ is the number of examples assigned to a class $k$, $n_k$ has a binomial distribution with probability of success $p = 1/L$ and $n$ trials. If $L \gg 1$, $\mathbb{E}|n_k - n_j| = \Theta(\sqrt{n/L})$ holds in expectation when $k \neq j$. On the other hand, $\mathbb{E}[n_k] = n/L$. Hence, the number of examples $n$ should satisfy $n \gg L$ for classes to be balanced. This is particularly important for demographic parity because the level of bias defined in (2) holds uniformly over *all* classes. While the formulation in (2) is independent of class imbalance, the estimate of $\epsilon$ might be less reliable if $n_k$ is small (i.e. class $k$ is rare) for some value of $k$.

# D  Proof of Proposition 1

First, we note that the optimization problem in (4) is convex. The objective function is quadratic on the optimization variable $\hat{y}$. In addition, the feasible set is convex because the maximum of affine functions is convex while the minimum of affine function is concave (Boyd and Vandenberghe, 2004). Hence:

$$\max_{s \in \mathcal{S}} \mathbb{E}_\mathbf{x}[\hat{y}_k(\mathbf{x}) \,|\, g(\mathbf{x}) = s] - \min_{s \in \mathcal{S}} \mathbb{E}_\mathbf{x}[\hat{y}_k(\mathbf{x}) \,|\, g(\mathbf{x}) = s]$$

is a sum of two convex functions on $\hat{y}$, therefore it is convex. Taking the maximum across $k$ labels retains convexity because the maximum of convex functions is convex (Boyd and Vandenberghe, 2004).

Let $\mathcal{C}$ be the set of feasible functions $\hat{y} : \mathcal{X} \to \mathbb{R}^L$ that satisfy the constraint: $\forall x \in \mathcal{D} : 0 \leq \hat{y}(x) \leq 1$ and the bias constraint in (2). Then, the optimization problem in (5) can be rewritten in the form:

$$\min_{\hat{y} \in \mathcal{C}} \quad \sum_{x \in \mathcal{D}} \left\{ (1/2) \|\hat{y}(x)\|_2^2 \;-\; \hat{y}(x)^T f(x) \;+\; \mathbb{I}_\mathcal{F}(z(x)) \right\}. \tag{7}$$

$$\text{s.t.} \quad \forall x \in \mathcal{D} : \; \hat{y}(x) = z(x)$$

The augmented Lagrangian using the scaled dual variables Boyd et al. (2011) is:

$$L_\tau(\hat{y}, z, u) = \sum_{x \in \mathcal{D}} \left\{ (1/2) \|\hat{y}(x)\|_2^2 - \hat{y}(x)^T f(x) + \mathbb{I}_\mathcal{F}(z(x)) + (\tau/2) \|\hat{y}(x) - z(x) + u(x)\|^2 \right\},$$

whose domain on its first argument is the feasible set $\hat{y} \in \mathcal{C}$. The scaled form of the ADMM updates are Boyd et al. (2011):

$$\hat{y}(x)^{(t+1)} = \arg\min_{\hat{y} \in \mathcal{C}} \sum_{x \in \mathcal{D}} \left\{ (1/2) \|\hat{y}(x)\|_2^2 \;-\; \hat{y}(x)^T f(x) \;+\; (\tau/2) \|\hat{y}(x) - z^{(t)}(x) + u^{(t)}(x)\|^2 \right\}$$

$$z(x)^{(t+1)} = \Pi_\mathcal{F}\{\hat{y}^{(t+1)}(x) + u^{(t)}(x)\}, \qquad u(x)^{(t+1)} = u(x)^{(t)} + \hat{y}(x)^{(t+1)} - z(x)^{(t+1)},$$

where $\Pi_\mathcal{F}(w(x))$ is the projection into the set $\{v \in \mathbb{R}^L : \mathbf{1}^T v = 1\}$. Using Lagrange duality, it can be shown that the solution to the latter projection problem has the closed-form:

$$z(x)^{(t+1)} = h^{(t+1)}(x) + u^{(t+1)}(x) - \mu(x)\mathbf{1}, \tag{8}$$

where:

$$\mu(x) = \frac{\mathbf{1}^T(h(x) + u(x)) - 1}{K}.$$

Denote $x_i$ for the $i$-th instance in the training dataset. Let $H^{(t)}, Z^{(t)}, U^{(t)} \in \mathbb{R}^{N \times L}$ be matrices, where $N$ is the number of training examples and $L$ is the number of classes, such that $H_{ik}^{(t)} = \hat{y}_k^{(t)}(x_i)$, $Z_{ik}^{(t)} = z_k^{(t)}(x_i)$, and $U_{ik}^{(t)} = u_k^{(t)}(x_i)$. Then, Algorithm 1 implements such ADMM updates in matrix form. Convergence and optimality are guaranteed by ADMM since the optimization problem is convex and contains a strongly-convex term Boyd et al. (2011); Nishihara et al. (2015).

# E  Proof of Proposition 2

Consider a single subpopulation in $\mathcal{X}_\mathcal{S}$ with $n_0$ training examples. We have by the triangle inequality:

$$\left|\mathbb{E}[f_k(\mathbf{x})] - \hat{\mathbb{E}}[h_k(\mathbf{x})]\right| \leq \left|\mathbb{E}[f_k(\mathbf{x})] - \hat{\mathbb{E}}[f_k(\mathbf{x})]\right| + \left|\hat{\mathbb{E}}[f_k(\mathbf{x})] - \hat{\mathbb{E}}[h_k(\mathbf{x})]\right|.$$

We consider the first term. First, if $f_k \in \mathcal{F}_k$ and $\mathcal{F}_k$ has a Rademacher complexity $d$, then with probability of at least $1 - \delta$, the following bounds holds uniformly for all functions $f_k \in \mathcal{F}_k$ Bousquet et al. (2003):

$$\left|\mathbb{E}[f_k(\mathbf{x})] - \hat{\mathbb{E}}[f_k(\mathbf{x})]\right| \leq 2d + \sqrt{\frac{\log \frac{2}{\delta}}{2n_0}}. \tag{9}$$

By the union bound, this holds for all the $L$ classes in $\mathcal{Y}$ and $R$ sub-populations in $\mathcal{S}$ with a probability of at least $1 - LR\delta$. Therefore, with a probability of at least $1 - \delta$, we have:

$$\sup_{k \in \mathcal{Y},\, s \in [R]} \sup_{f_k \in \mathcal{F}_k} \left|\mathbb{E}[f_k(\mathbf{x}) \mid g(\mathbf{x}) = s] - \hat{\mathbb{E}}[f_k(\mathbf{x}) \mid g(\mathbf{x}) = s]\right| \leq 2d + \sqrt{\frac{\log \frac{2LR}{\delta}}{2n_0}}. \tag{10}$$

On the other hand:

$$
\begin{aligned}
DP(f) &= \sup_{k \in \mathcal{Y}} \left\{ \max_{s \in [R]} \mathbb{E}[f_k(\mathbf{x}) \mid g(\mathbf{x}) = s] - \min_{s \in [R]} \mathbb{E}[f_k(\mathbf{x}) \mid g(\mathbf{x}) = s] \right\} \\
&\leq \sup_{k \in \mathcal{Y}} \left\{ \max_{s \in [R]} \hat{\mathbb{E}}[h_k(\mathbf{x}) \mid g(\mathbf{x}) = s] - \min_{s \in [R]} \hat{\mathbb{E}}[h_k(\mathbf{x}) \mid g(\mathbf{x}) = s] \right\} \\
&\quad + 2 \sup_{k \in \mathcal{Y},\, s \in [R]} \sup_{f_k \in \mathcal{F}_k} \left|\mathbb{E}[f_k(\mathbf{x}) \mid g(\mathbf{x}) = s] - \hat{\mathbb{E}}[h_k(\mathbf{x}) \mid g(\mathbf{x}) = s]\right| \\
&\leq \epsilon + 2 \sup_{k \in \mathcal{Y},\, s \in [R]} \sup_{f_k \in \mathcal{F}_k} \left|\mathbb{E}[f_k(\mathbf{x}) \mid g(\mathbf{x}) = s] - \hat{\mathbb{E}}[h_k(\mathbf{x}) \mid g(\mathbf{x}) = s]\right|
\end{aligned}
$$

Here, we used the fact that $h(\mathbf{x})$ is debiased using Algorithm 1 so it satisfies demographic parity on the training examples with level $\epsilon$ (by Proposition 1). Therefore, using (10), we have with a probability of at least $1 - \delta$:

$$
\begin{aligned}
DP(f) &\leq \epsilon + 2 \sup_{k \in \mathcal{Y},\, s \in [R]} \left|\hat{\mathbb{E}}[f_k(\mathbf{x}) \mid g(\mathbf{x}) = s] - \hat{\mathbb{E}}[h_k(\mathbf{x}) \mid g(\mathbf{x}) = s]\right| + 2d + \sqrt{\frac{\log \frac{2LR}{\delta}}{2n_0}} \\
&= \epsilon + 2\tau + 2d + \sqrt{\frac{\log \frac{2LR}{\delta}}{2n_0}}.
\end{aligned}
$$

# F  Top-K Results for Adult Income Dataset

See Table 5.

Table 5: A summary of the top-2 and top-3 results (99% confidence intervals) for the five debiasing algorithms on the Adult Income benchmark dataset, where the goal is to predict the education level. See Table 1 for further details about the experiment setup.

| $p$ | | BL | | DPR | | OST | |
|---|---|---|---|---|---|---|---|
| | | TOP-2 [%] | TOP-3 [%] | TOP-2 [%] | TOP-3 [%] | TOP-2 [%] | TOP-3 [%] |
| 0.0 | RF | $65.3 \pm 0.5$ | $77.1 \pm 0.3$ | $63.6 \pm 0.5$ | $75.6 \pm 0.3$ | $65.3 \pm 0.5$ | $77.1 \pm 0.3$ |
| | KNN | $52.7 \pm 0.4$ | $64.4 \pm 0.5$ | $49.8 \pm 0.4$ | $62.1 \pm 0.2$ | $52.2 \pm 0.4$ | $64.0 \pm 0.5$ |
| 0.5 | RF | $63.4 \pm 0.4$ | $76.6 \pm 0.3$ | $59.5 \pm 0.4$ | $72.8 \pm 0.5$ | $55.4 \pm 0.6$ | $71.6 \pm 0.4$ |
| | KNN | $51.6 \pm 0.5$ | $62.5 \pm 0.5$ | $49.2 \pm 0.5$ | $60.1 \pm 0.5$ | $41.4 \pm 9.5$ | $53.4 \pm 11.8$ |
| $p$ | | HTO | | R2B$_0$ | | R2B (ours) | |
| | | TOP-2 [%] | TOP-3 [%] | TOP-2 [%] | TOP-3 [%] | TOP-2 [%] | TOP-3 [%] |
| 0.0 | RF | $65.2 \pm 0.6$ | $76.9 \pm 0.3$ | $65.1 \pm 0.6$ | $76.8 \pm 0.3$ | $65.0 \pm 0.6$ | $76.8 \pm 0.3$ |
| | KNN | $52.1 \pm 0.4$ | $63.9 \pm 0.6$ | $52.1 \pm 0.4$ | $63.9 \pm 0.6$ | $51.9 \pm 0.4$ | $63.7 \pm 0.4$ |
| 0.5 | RF | $55.1 \pm 0.6$ | $71.8 \pm 0.5$ | $54.9 \pm 1.0$ | $71.7 \pm 0.7$ | $55.0 \pm 0.3$ | $66.0 \pm 0.4$ |
| | KNN | $44.0 \pm 0.6$ | $56.9 \pm 0.5$ | $44.0 \pm 0.5$ | $57.0 \pm 0.5$ | $44.4 \pm 0.8$ | $57.6 \pm 0.8$ |

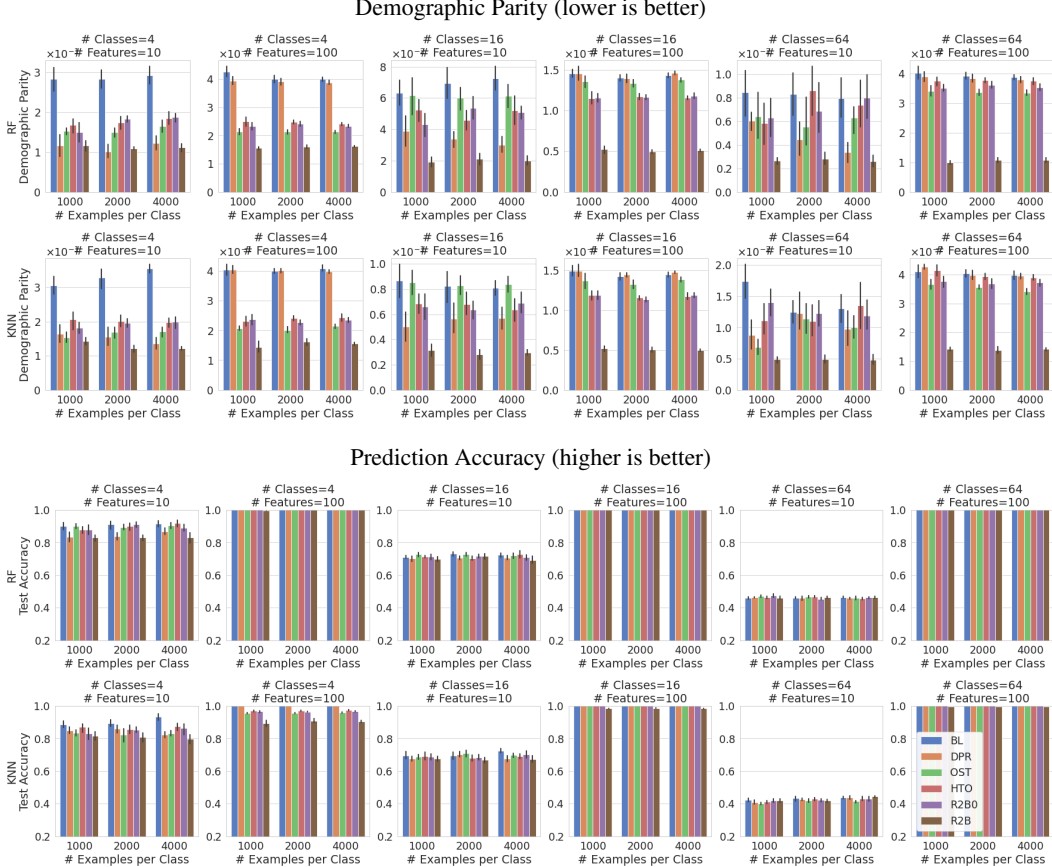

Figure 4: The top two rows display the multiclass demographic parity as measured using (2) for the synthetic dataset with varying numbers of classes, features, and training examples. The first row is for the Random Forest (RF) classifier while the second row is for $k$-NN. Each color represents a debiasing technique. The reduction-to-binary (R2B) method provides a stronger fairness guarantee than the competing methods. In addition, we observe that transforming the features performs poorly when the number of features is large (see the discussion in Section 4). The bottom two rows show the prediction accuracy in each case. Note that the drop in accuracy in the case of 64 classes with 10 features is expected because the 64 classes have a large overlap in that setting.

## G    Random Forests Classifier Figures on Synthetic Data

In addition to the $k$-NN figures for the synthetic data experiment, we also provide here similar results using the random forests classifiers. See Figures 4 and 5.

## H    Training Procedure for COCO Dataset.

We train a linear classifier with softmax activations and cross entropy loss using the 2,048 pre-logit features of a ResNet50 backbone (He et al., 2016) pretrained on ImageNet-ILSVRC2012 (Deng et al., 2009). We use SGD with momentum 0.9 for 400 epochs and the following learning rate schedule: $10^{-3}$ (200 epochs), $10^{-4}$ (100 epochs) and $10^{-5}$ (100 epochs).

## I    Dermatology ethics approval and data availability

The images and metadata were de-identified according to Health Insurance Portability and Accountability Act (HIPAA) Safe Harbor prior to transfer to study investigators. The protocol was reviewed

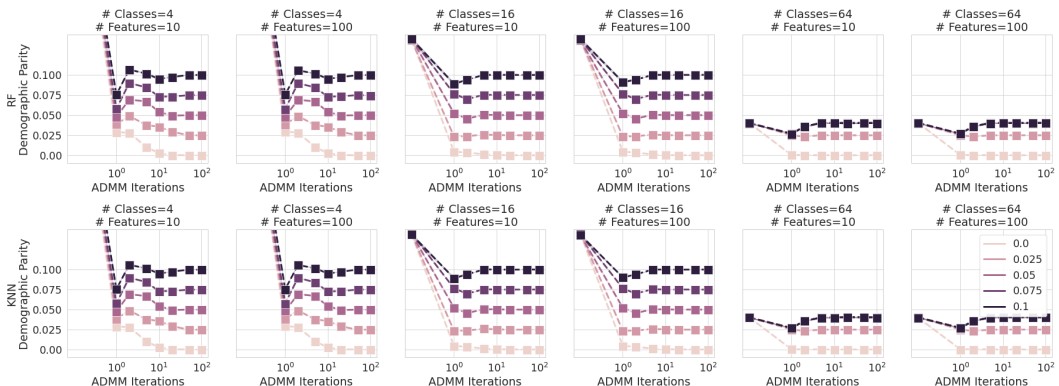

Figure 5: The level of bias in the *training* data is plotted as measured by (2) vs. the number of ADMM rounds for different levels of bias tolerance $\epsilon \in \{0, 0.025, 0.05, 0.075, 0.10\}$ (see legend). In general, 50-100 rounds of ADMM are sufficient to reach convergence in agreement with prior observations Boyd et al. (2011). In addition, the level of bias in the training data indeed converges to the minimum of the original data bias level and the prescribed tolerance level $\epsilon$. The number of ADMM rounds in each figure is varied in the set $\{1, 10, 20, 50, 100\}$, where the leftmost point corresponds to original bias in the data.

by [Anonymous] IRB, which determined that it was exempt from further review under 45 CFR 46. The dermatology data is not available to the public.

## J  Running Time Analysis

The reduction-to-binary (R2B) algorithm is a lightweight preprocessing method because it operates on the labels only. In Figure 5, we show that ADMM converges quickly in a few rounds in agreement with earlier observations in the literature (Boyd et al., 2011). The time complexity for each single label is $O(|\mathcal{S}|)$ in each round, where $\mathcal{S}$ is the set of demographic groups (Alabdulmohsin and Lucic, 2021). Within each group, the required running time can be controlled according to the desired level of accuracy. For example, using a batch size of 256 and a total of 100K SGD steps, training on a single NVIDIA V100 Tensor Core GPU takes approximately 5s per label in each round of R2B. However, labels can be debiased in parallel in each round of the R2B algorithm.