# OpenReview forum: "A Reduction to Binary Approach for Debiasing Multiclass Datasets"
_NeurIPS.cc/2022/Conference — NeurIPS 2022 Accept_

### Official Review · Reviewer_jQyz · 2022-07-08

**Rating:** 6
**Confidence:** 3
**Soundness:** 3 good
**Presentation:** 3 good
**Contribution:** 3 good

**Summary:**

The paper proposed a novel reduction-to-binary (R2B) approach that enforces demographic parity for multiclass classification with non-binary sensitive attributes via a reduction to a sequence of binary debiasing tasks. It proved that R2B satisfies optimality and bias guarantees. It also demonstrated empirically that it can lead to improvements over baselines and produced competitive results. The authors validated the method on synthetic and real-world datasets from social science, computer vision, and healthcare.

**Questions:**

Overall, this is a good paper with solid experiments and analysis, and I can see its potential for the broader field.

Please see the weakness part for detailed questions. I paste here some key questions for summary

1. No fine-grained results provided for R2B on where the improvements mainly come from.

2. The datasets evaluated were relatively small-scale.

3. No runtime / training cost analysis.
4. The overall performance was comparable to other algorithms, but no clear advantage.

**Limitations:**

The authors have adequately addressed the limitations in the last section of the main paper.

**Strengths And Weaknesses:**

### Strengths

1. The paper studies an interesting yet important topic, namely multiclass classification with non-binary sensitive attributes.

2. The paper proposed a debiasing method for multiclass datasets with categorical sensitive attributes. It further established theoretical guarantees for the proposed algorithm.

3. The evaluation was done on real-world datasets from social science, computer vision, and healthcare domains. The performance is notable compared to competing baselines.

4. The writing is clear and easy to follow. The paper is well structured.

### Weaknesses
1. Where does the improvement come from? As the authors noted, potential bias and class imbalance could cause existing algorithms fail for underrepresented data. Yet, no fine-grained results provided for R2B on where the improvements mainly come from. For example, for classes with fewer observations, does it produce more balanced / fair predictions? If so, why is it the case?

2. The datasets evaluated were relatively small-scale. The number of samples and data diversity seem to be limited for the datasets tested. What would be the case when you scale the dataset size up? Without validation on larger datasets, the performance may not be justifiable.

3. No runtime / training cost analysis. What is the additional cost (both time and memory) when introduing the debias procedure?

4. The overall performance was comparable to other algorithms, but no clear advantage. Some of the numbers seem to lie in the noise region.

---

> ### Author Response · Authors · 2022-07-27
> **Authors' Response**
>
> Thank you for your comments. Please see our response below. We hope this clarifies your main concerns.
>
> - *"Where does the improvement come from?"*:
> We discuss this briefly in the paper. For example, the feature transformation approach requires an accurate estimate of the CDFs (Lines 32-39). In Appendix B, we demonstrate how this can fail at debiasing tasks. In label transformation methods, normalization plays an important rule as we discuss in Lines 28-31 and Figure 1. R2B yields an optimal solution subject to the normalization constraints because it takes it explicitly into account.
>
> - *Small-scale datasets*: Please see our response to Reviewer DJH3.
>
> - *No runtime / training cost analysis*: The algorithm is lightweight because it operates on the labels only. It’s negligible compared to the cost of training models, particularly deep neural networks. In Figure 3, we show that ADMM converges quickly in a few rounds. In each round, RTO converges quickly (see Proposition 1 in Alabdulmohsin & Lucic, 2021).
>
> - *Performance*: In Dermatology, the performance of R2B is comparable to OST. However, R2B performs better than the other methods in all the other cases (Adult dataset, COCO, and the synthetic data experiments).

---

> > ### Author Response · Authors · 2022-07-29
> > **Follow up**
> >
> > Dear reviewer,
> >
> > Thank you again for the comments. We have updated the paper to reflect the comments. We also included a section in the appendix about the runtime / training cost.
> >
> > We hope this addresses your concerns. If there are remaining concerns, we would appreciate it if you could let us know so that we can respond to them during the discussion.

---

### Official Review · Reviewer_EEH6 · 2022-07-08

**Rating:** 5
**Confidence:** 2
**Soundness:** 2 fair
**Presentation:** 2 fair
**Contribution:** 2 fair

**Summary:**

The paper considers the fairness of the multi-class classification task. The authors reduce a multi-class debasing problem to a set of parallel binary classification debasing tasks for each class label k, followed by an aggregation step. For the binary debias task, they leverage existing method e.g., RTO, to achieve the debiased goal for each label. They apply their debias technique in the pre-processing step: with the "debiased" labels obtained from their R2B method, they compare with some other pre-processing methods in terms of the DP fairness goal and model accuracy.

**Questions:**

1. As indicated above, is there a way to verify why the normalization method in Alg.1 will not re-introduce bias (or reduce that)?
2. For the comparison of the results, e.g, in Fig.2, the DP is in $10^{-3}$ order, then how much difference can we say between your method with other baselines? Is such a difference significant?
3. For the comparison with other baselines, e.g., with the HTO, what if you did not set the $\gamma$ as the hard-thresholding but follow Alabudulmohsin 2021 to do the grid search? Will your R2B still outperform it?
4. Others:
    - The y-axis in Fig.3 is not correct. The maximum value should be 0.1 (or *1/10 for other values).
    - It will be great if you can introduce a concept before using it in the paper. For example, in the introduction, the authors mentioned "label debiasing" or "debiased label" without defining it, which makes it hard to follow.
 - The $**2$ in $l_2$ distance in #149 seems in the wrong position.
 - #155: do you mean correctly classifying or misclassifying?

**Limitations:**

Yes

**Strengths And Weaknesses:**

Existing literature about fairness usually focuses on the binary classification task, this paper extends this to the multi-class scenario which is a more realistic setting and provides a pre-processing methodology to do this. With their reduction, the algorithm can scale to large datasets.

While their motivation is important and realistic, it seems to me, the authors just reduce the multiclass problem to a set of binary cases and do an aggregation without further verifying why such an aggregation method works. The aggregation method seems to matter a lot in their method as indicated in the introduction, the normalization could re-introduce bias. However the authors do not mention how their choice can address this re-introducing issue. Also, for the experimental results, it is hard for readers to distinguish between a DP score 5.1 vs 5.0 (e.g., Table 3).

---

> ### Author Response · Authors · 2022-07-27
> **Authors' Response**
>
> Thank you for your comments. Please see our response below. We hope this clarifies your main concerns.
>
> - *Aggregation Rule*:
> The aggregation method we propose is not ad-hoc; it is derived from the ADMM procedure and we *prove* that it recovers the optimal solution. In addition, we compare against the baseline aggregation rule and show empirically that our algorithm yields better results.
>
> - *"How does the aggregation method handle normalization?"*:
> We prove that our aggregation method recovers the optimal solution subject to the normalization constraint. Please note that Algorithm 1 produces the optimal solution to (5), where we add the indicator function of the feasible set $\hat y^T1=1$  into the objective function. In particular, it is easy to show that the solution $\hat Y$ in Algorithm 1 is indeed normalized (cf. the Equation for Z).
>
> - *"it is hard for readers to distinguish between a DP score 5.1 vs 5.0 (e.g., Table 3)."*:
> True. That’s why we include error bars and both results are highlighted in boldface because they are within a margin of error of each other. We don't claim that 5.0 is better than 5.1 in Table 3.
>
> - *DP in Figure 2*:
> In Figure 2, DP is of the order $10^{-2}$ in 5 out of the 6 cases. But, even in the case where DP is of the order $10^{-3}$, the difference between methods is statistically significant. Please note that we report the error bar in the figure.
>
> - *Hard thresholding rule (HTO)*:
> We have observed that optimizing gamma gives a performance similar to OST, which is another soft-thresholding rule. R2B continues to perform better. We report results for the hard-thresholding rule because it is quite different from OST and R2B0 so it would be more informative.
>
> - *Other comments*:
> Thank you for pointing out these typos. We will fix them.

---

> > ### Author Response · Authors · 2022-07-29
> > **Follow up**
> >
> > Dear reviewer,
> >
> > Thank you again for the comments. We will update the paper to correct the typos and clarify the points you highlighted (e.g. how R2B handles normalization and why its aggregation rule works). Please note that we have verified the optimality of R2B by comparing its solution with the optimal solution to (4) produced by the solver CVX [1].
> >
> > We hope this addresses your concerns. If so, we would appreciate it if you would consider revising your score or let us know if there are still remaining issues unaddressed so that we can respond to them during the discussion.
> >
> > [1] Michael Grant and Stephen Boyd. CVX: Matlab software for disciplined convex programming, version 2.0 beta. http://cvxr.com/cvx, September 2013.

---

### Official Review · Reviewer_qxof · 2022-07-10

**Rating:** 6
**Confidence:** 3
**Soundness:** 2 fair
**Presentation:** 3 good
**Contribution:** 3 good

**Summary:**

The authors introduced a method to extend the current debiasing methods to the multi-class classification problems. The core idea is to formulate the loss function as an L2 regression loss constraint by traditional DP.  Then the optimization problem is solved by ADMM algorithm efficiently; each iteration is only required to solve a binary classification subproblem. The optimality and bound of bias are analyzed. Finally, experiments on both synthetic and commonly used real datasets are conducted to show the reduction in bias while accuracy is sustained.


**Questions:**

The first limitation of this paper seems to be the loss function itself. Authors choose L2 regression loss instead of the more commonly used cross-entropy loss. This assumption seems critical for applying ADMM.

Furthermore, I doubt the correctness of the optimality theorem in the paper. It is true that the whole function is strongly convex as a function of yhat, but we are really optimizing over the model parameters in yhat. Even if we consider yhat as free variables, they are constrained by nonconvex set C = DP <= \epsilon.

There are a few places that are confusing to me:

L140 “Minimizing the expected 0-1 error is equivalent to minimizing the linear…”: I think it should be maximizing the linear functional rather than minimizing.

L148: “the squared loss is statistically consistent”, I checked (Yang et al. 2020), but I can’t find a relevant claim that squared loss is statistically consistent.

L155: “probability of misclassifying an example x equals y^(x)T y(x)”: this should be correctly classifying the example x?

Finally, the format of the tables is ambiguous: the boldface is applied to some (but not all) best methods. For example, in Table 2, only R2B column is highlighted, but other methods are better in DP or accuracy. Similar issue is found in Table 3. In Table 4 there is no boldface in the whole table. So I think efforts are needed to make it more consistent.


**Limitations:**

Please see the section above.

**Strengths And Weaknesses:**

This paper is well organized and smooth to read in general, although I found some unclear places (listed in the next section). The idea of transforming the multi-class problem into multiple binary problems is novel and elegant to me. The empirical evaluations are sufficient.

---

> ### Author Response · Authors · 2022-07-27
> **Authors' Response**
>
> Thank you for your comments. Please see our response below. We hope this clarifies your main concerns.
>
> - *L2 regression loss*: We don’t believe this qualifies as a "limitation" since our algorithm treats classifiers as black-box methods. During training, the cross-entropy loss can be used if desired after preprocessing the data. The use of the L2 loss is part of the design of the preprocessing algorithm; it leads to the desired decomposition across labels and it is top-K consistent.
>
> - *"I doubt the correctness of the optimality theorem in the paper"*:
> The set C = DP <= \epsilon is convex. The DP criterion is a composition of a convex function with an affine function, hence it is convex. The feasible set in the constraint DP <= \epsilon is convex because it’s a sublevel set of a convex function (cf. https://web.stanford.edu/~boyd/cvxbook/). We have elaborated more on this point in the revised version in Lines 532-537.
>
> - *Typos in L140 and L155*: Yes, these are typos. We’ll fix them. Thank you for pointing them out.
>
> - *Consistency for top-K loss*: Please see Section 3 in Yang, F. and Koyejo, S. (2020). The square loss is a Bregman divergence.
>
> - *Boldface in tables*: We will clarify this in the paper. First, we identify the best methods in terms of bias within the margin of error. If multiple methods perform equally well in terms of DP, we highlight the ones that have the highest accuracy (again within the margin of error). In Table 2, for example, there aren’t any methods better than R2B in terms of DP because OST performs equally well (within margin of error). However, R2B performs better than OST in terms of accuracy so it’s identified as the best method. We have clarified this in the revised version (Lines 209-210).

---

> > ### Author Response · Authors · 2022-07-29
> > **Follow up**
> >
> > Dear reviewer,
> >
> > Thank you again for the comments. We will update the paper to correct the typos and clarify the points you highlighted (i.e. about the convexity of the feasible set and the use of boldface in the tables). The proof of optimality is correct and we have verified this by comparing the solution with the optimal solution to (4) produced by the solver CVX [1].
> >
> > We hope this addresses your concerns. If so, we would appreciate it if you would consider revising your score or let us know if there are still remaining issues unaddressed so that we can respond to them during the discussion.
> >
> > [1] Michael Grant and Stephen Boyd. CVX: Matlab software for disciplined convex programming, version 2.0 beta. http://cvxr.com/cvx, September 2013.

---

### Official Review · Reviewer_DJH3 · 2022-07-12

**Rating:** 4
**Confidence:** 4
**Soundness:** 2 fair
**Presentation:** 3 good
**Contribution:** 2 fair

**Summary:**

This paper proposes a reduction-to-binary method, which enforce demographic parity for multi-class classification with non-binary sensitive attributes. It mainly exploits the alternating direction method of the multipliers (ADMM) to decompose the task of debiasing multi-class datasets into a sequence of parallel debiasing jobs. Experiments are implemented on several datasets to demonstrate its effectiveness.

**Questions:**

Please see the weakness part for the novelty, fairness, and experimental comparison in the above comments.

**Limitations:**

Yes.

**Strengths And Weaknesses:**

Strengths：
This paper introduces a pre-processing method for multi-class classification with non-binary sensitive attributes with ADMM and experiments are performed to show the performance. In addition, the algorithm are analyzed from different theoretical perspectives.

Weaknesses:
First, the novelty is relatively limited. It mainly extends RTO [1] with ADMM and apply it as a pre-processing method. For example, [2] also exploits ADMM to optimize score transformation for fair classification.

Second, this method is pre-processing method for debiasing multiclass datasets with categorical sensitive attributes. However, some problems are ignored. On the one hand, since the method builds upon the narrow definition of demographic parity, it may not ensure the true fairness [3]. Though this has been discussed in Sec.5 of the paper, fairness is an important motivation for such debiasing work as illustrated in the introduction part. More criterion should be considered. On the other hand, the proposed R2B method does not consider the influences of noisy labels when performing label debiasing.

Further, the experiments are not enough and training settings are relatively simple. For COCO dataset, linear classifiers with softmax activations and cross entropy loss are trained using pretrained features. The total number of examples is 22,616. Such way of training is difficult to demonstrate the effectivity of the proposed method in large-scale datasets when training from scratch is required. As a pre-processing method, more representative models on large-scale datasets should be re-trained to verify the performance.

[1] Alabdulmohsin I M, Lucic M. A near-optimal algorithm for debiasing trained machine learning models[J]. Advances in Neural Information Processing Systems, 2021, 34: 8072-8084.

[2] Wei D, Ramamurthy K N, Calmon F P. Optimized score transformation for fair classification[J]. Proceedings of Machine Learning Research, 2020, 108.

[3] Wang Z, Qinami K, Karakozis I C, et al. Towards fairness in visual recognition: Effective strategies for bias mitigation[C]//Proceedings of the IEEE/CVF conference on computer vision and pattern recognition. 2020: 8919-8928.

---

> ### Author Response · Authors · 2022-07-27
> **Authors' Response**
>
> Thank you for your comments. Please see our response below. We hope this clarifies your main concerns.
>
>
> - *Novelty*:
> Our contribution is not only to extend RTO, but also to develop a “reduction” approach where RTO can be applied to each label separately while still establishing solid theoretical guarantees. Hence, even when the number of classes is large, labels can be debiased in parallel in each round. We show that ADMM can be used to achieve that goal and that the aggregation method guarantees recovering the optimal solution.
>
> - *Reference [2] is missing.*: It is not missing! It is both discussed in the related works section and we also include it in the evaluation in all experiments (denoted as OST).
>
> - *Demographic Parity vs. other metics*: We do acknowledge in the paper that DP is only one technical definition that doesn’t capture fairness in its broader sense. But, it remains an important and widely adopted definition nevertheless. Please see our response to Reviewer tbyd above.
>
> - *Experiments are not enough*:
> We have conducted the evaluation in three domains: social science, computer vision, and healthcare. We also used different classifiers, e.g. kNN, random forests, and deep neural networks. In the latter case, pre-training is a standard approach to achieve state-of-the-art results. We also include experiments on synthetic data where we vary the number of classes, groups, dimensionality, and examples. If there are specific datasets related to ML fairness with more examples than 50K, we would appreciate it if you could point them out.

---

> > ### Author Response · Authors · 2022-07-29
> > **Follow up**
> >
> > Dear reviewer,
> >
> > Thank you again for the comments. We have updated the paper to reflect the comments. We hope this addresses your concerns. If so, we would appreciate it if you would consider revising your score or let us know if there are still remaining issues unaddressed so that we can respond to them during the discussion.

---

> > ### Comment · Reviewer_DJH3 · 2022-08-10
> > **Thanks for the response**
> >
> > The review has been updated accordingly. The authors have addressed some concerns. However, I still think it has some limitations on the used DP metric and the corresponding evaluation part.

---

### Official Review · Reviewer_tbyd · 2022-07-12

**Rating:** 5
**Confidence:** 3
**Soundness:** 3 good
**Presentation:** 3 good
**Contribution:** 3 good

**Summary:**

This paper tries to debias multi-classification models. The proposed method builds on top of the alternating direction method of the multipliers algorithm. It decomposes debiasing multi-classification into a sequence of parallel jobs on each class, followed with an aggregation step. The paper includes experimental results on both synthetic and real-world datasets and found the resultspromising.

**Questions:**

Discussed in "Strengths and Weakness" section.

**Limitations:**

The paper does emphasize that "fairness" is a societal concept and can not be reduced to statistical metrics. It also discussed the limitations of the proposed method and gives reasonable explanations and potential solutions.

**Strengths And Weaknesses:**

Strengths:
1. A new way of debiasing multi-classification models.
2. The paper provides both theoretical guarantees and empirical results.
3. The paper explores both classic machine learning models and deep learning models.


Weakness:
1. While we do see that demographic parity is reduced by R2B(the proposed method), the test accuracy also drops, especially when the number of classes is relatively small. This is captured in Figure 2 and Table 1.
2. As pointed out by the paper itself, error parity increases when the dataset has large bias.
3. The paper may benefit from having analysis on how the proposed method perform differently when models are different (from different families of machine learning models).

---

> ### Author Response · Authors · 2022-07-27
> **Authors' Response**
>
> Thank you for your comments. Please see our response below. We hope this clarifies your main concerns.
>
> - *Impact on accuracy*:
> This is to be expected. If the classes are correlated with the sensitive attribute, debiasing the model will impact its accuracy. The existence of this tradeoff has been pointed out in several works, e.g.  [1, 2, 3]. We have added an acknowledgement of this in the revised version in the Limitations section (please see Lines 308-310) .
>
> - *Error parity*:
> The focus in this paper is on demographic parity (DP), which is one of the most commonly used statistical measures of bias. Several recent works have pointed out an impossibility to satisfy multiple notions of bias simultaneously [4, 5], so we do expect controlling for DP to impact error parity in some cases. However, we show in Table 4 that the impact is small unless the bias in the original dataset was quite significant (i.e. $DP > 0.5$). Our focus is on cases where the most suitable notion of bias is DP, e.g. in hiring and credit scoring, where the predictions of the model should be independent of the demographic information. Also, please note that DP captures most closely the common perception of bias (Srivastava et al., 2019). In addition, it arises naturally in causal predictive tasks (Veitch et al. 2021). These issues are discussed in the paper; e.g. in Section 1, 3.1, and 5. For all these reasons, we believe that developing algorithms to mitigate DP has its merits. This is especially true given that we handle the broad setting of multiclass classification with non-binary sensitive attributes, which is the norm in practice, but for which very few methods exist.
>
> [1] Menon, A. K. and Williamson, R. C. The cost of fairness in binary classification. In FaaCT, pp. 107-118, 2018.
>
> [2] Chen, I. Y., Johansson, F. D., and Sontag, D. Why is my classifier discriminatory? NeurIPS, 2018.
>
> [3] Zhao, H. and Gordon, G. J. Inherent tradeoffs in learning fair representation. arXiv preprint arXiv:1906.08386, 2019
>
> [4] A. Chouldechova, “Fair prediction with disparate impact: A study of bias in recidivism prediction instruments,” Big data, vol. 5, no. 2, pp. 153–163, 2017.
>
> [5] J. Kleinberg, S. Mullainathan, and M. Raghavan, “Inherent trade-offs in the fair determination
> of risk scores,” 2017.

---

> > ### Author Response · Authors · 2022-07-29
> > **Follow up**
> >
> > Dear reviewer,
> >
> > Regarding the third point you mentioned about the analysis on different ML model families, we have applied the method on different families of ML models, including kNN, random forests, and deep neural networks. We do not observe any patterns in the results; the proposed algorithm works well in all cases. If there is a specific type of analysis you are looking for, please let us know.
> >
> > We hope we have addressed your concerns in our response above. If so, we would appreciate it if you would consider revising your score or let us know if there are still remaining issues unaddressed so that we can respond to them during the discussion.

---

### Author Response · Authors · 2022-07-29
**Summary of changes**

We thank the reviewers for the valuable comments. We appreciate the positive feedback on the novelty of the method, the importance of the topic (multiclass classification with non-binary sensitive attributes), the theoretical guarantees, and the clarity of writing.

We have improved the paper further by incorporating the suggestions made by the reviewers. Below is a summary of the changes:

- Added a proof about the convexity of the feasible set as suggested by Reviewer qxof (Lines 532-537). Please note that we have verified that R2B produces the optimal solution by comparing it with the solution produced by the solver CVX [1].

- Added a discussion and references about the tradeoff between bias and accuracy (Lines 308-310) as highlighted by reviewer tbyd.

- Included an explanation for the use of boldface in the tables, as requested by Reviewer qxof (Lines 209-210).

- Added to the introduction a definition of debiasing labels (Lines 28-29) as suggested by Reviewer EEH6.

- Fixed the typos, including the typo in the y-axis of Figure 3.

- Included a section in the appendix about the running time of the algorithm, as requested by Reviewer jQyz.

We hope this addresses your comments. If not, please let us know about any remaining concerns so we can address them.

[1] Michael Grant and Stephen Boyd. CVX: Matlab software for disciplined convex programming, version 2.0 beta. http://cvxr.com/cvx, September 2013.

---

### Comment · Area_Chair_yVSK · 2022-08-07
**Reviewer-author discussion post rebuttal**

Dear Reviewers,

Thank-you all for your work on reviewing this paper. The authors have responded to your comments in great detail.

Can you please read the response for each of your reviews and do the following?
1. Let the authors' know what parts of the response are satisfactory or not satisfactory (stating the reason).
2. If needed, ask followup questions.
3. If appropriate, increase the score.

Since the discussion period is ending soon, please engage with the authors asap.

Thanks,
Your AC

---

### Meta-Review · Area_Chair_yVSK · 2022-08-30

**Recommendation:** Accept
**Confidence:** Less certain

**Metareview:**

This paper considers the problem of debiasing multiclass datasets using a reduction-to-binary (R2B) approach for demographic parity as a pre-processing step. This reduction is solved using a ADMM-style algorithm. It offers theoretical analysis of algorithm under some conditions. Experimental results on three datasets show reasonably good results.

All reviewers appreciated the reduction formulation to solve the less-studied problem of debiasing multi-class datasets. They also raised a number of concerns including the narrow problem setting, aggregation of results from reduction, and empirical results. Authors' have responded to those questions/concerns, but only one reviewer followed up and the rest did not.

Based on my own quick reading of the paper and from reviewer comments, I think the paper needs improvement on the following aspects.
1. The reduction approach is somewhat specific. There is a natural strategy to extend Agarwal et al. (2018)'s approach to multiclass, using the multiclass-to-binary reductions pioneered by Langford et al. (see https://hunch.net/~jl/projects/reductions/reductions.html) This is out of scope of the current paper, but should be mentioned for literature review reasons. The potential benefit is that it can be a sequential reduction that takes into consideration where the classifier is doing well or poorly.
2. As one of the reviewer mentioned, aggregation of results is critical step in reduction style approaches. There is not much detail / intuitive explanation for this in the paper (Line 165-169 doesn't have enough details).
3. Experiments are mostly focused on tabular datasets and non-parameteric classifiers (e.g., Random Forest and k-NN). There is one setting (COCO dataset) where a linear classifer is trained on pre-trained features. One reviewer asked the following question which gets to the core empirical question: "Where does the improvement come from? As the authors noted, potential bias and class imbalance could cause existing algorithms fail for underrepresented data. Yet, no fine-grained results provided for R2B on where the improvements mainly come from. For example, for classes with fewer observations, does it produce more balanced / fair predictions? If so, why is it the case?" I didn't find the authors' response satisfactory. I encourage the authors' to address this aspect if the paper is accepted.
4. The theory which is mostly straightforward states that classifier trained on debiased data is guaranteed to exhibit small bias in future as long as four conditions are met. Two relevant ones for experiments are: a) level of bias in the training data is small, b) there exists a large number of training examples for each group. The paper should have done some ablation experiments to demonstrate what happens when these conditions are not met and how much the results change as these conditions are relaxed.

For all the above reasons, I think the paper is borderline. I'm leaning towards accepting. If the paper is accepted, I strongly encourage the authors' to incorporate the above feedback in the final paper.

**Award:**

No

---

### Decision · Program_Chairs · 2022-09-14

Accept